# Single-cell analysis supports a luminal-neuroendocrine transdifferentiation in human prostate cancer

Baijun Dong [1,4], Juju Miao[2,3,4], Yanqing Wang[1,4], Wenqin Luo[2], Zhongzhong Ji [2], Huadong Lai[2,3], Man Zhang[2,3], Xiaomu Cheng[2,3], Jinming Wang[1], Yuxiang Fang[1,2], Helen He Zhu[1,2], Chee Wai Chua [1,2], Liancheng Fan[1], Yinjie Zhu[1], Jiahua Pan[1], Jia Wang[1,2✉], Wei Xue[1✉] & Wei-Qiang Gao[2,3✉]

Neuroendocrine prostate cancer is one of the most aggressive subtypes of prostate tumor. Although much progress has been made in understanding the development of neuroendocrine prostate cancer, the cellular architecture associated with neuroendocrine differentiation in human prostate cancer remain incompletely understood. Here, we use single-cell RNA sequencing to profile the transcriptomes of 21,292 cells from needle biopsies of 6 castration-resistant prostate cancers. Our analyses reveal that all neuroendocrine tumor cells display a luminal-like epithelial phenotype. In particular, lineage trajectory analysis suggests that focal neuroendocrine differentiation exclusively originate from luminal-like malignant cells rather than basal compartment. Further tissue microarray analysis validates the generality of the luminal phenotype of neuroendocrine cells. Moreover, we uncover neuroendocrine differentiation-associated gene signatures that may help us to further explore other intrinsic molecular mechanisms deriving neuroendocrine prostate cancer. In summary, our single-cell study provides direct evidence into the cellular states underlying neuroendocrine transdifferentiation in human prostate cancer.

[1] Department of Urology, Renji Hospital, School of Medicine, Shanghai Jiao Tong University, Shanghai 200127, China. [2] State Key Laboratory of Oncogenes and Related Genes, Renji-Med-X Stem Cell Research Center, Department of Urology, Ren Ji Hospital, School of Medicine and School of Biomedical Engineering, Shanghai Jiao Tong University, Shanghai 200127, China. [3] School of Biomedical Engineering and Med-X Research Institute, Shanghai Jiao Tong University, Shanghai 200030, China. [4]These authors contributed equally: Baijun Dong, Juju Miao, Yanqing Wang. ✉email: wj860520@163.com; xuewei@renji.com; gao.weiqiang@sjtu.edu.cn

Lineage plasticity endows cancer cells with the ability to switch their cellular phenotype[1] and is often associated with more aggressive stages of cancers[2]. In prostate cancer, lineage plasticity contributes to the acquisition of the neuroendocrine (NE) phenotype[3–5], with the emergence of a highly aggressive variant, termed neuroendocrine prostate cancer (NEPC)[6]. Current studies support that NEPC tumors arise clonally from prostate adenocarcinoma (PCA)[7], accompanying with a phenotypic transition from acini epithelial tumor cells to NE-like tumor cells[8]. This lineage transition enables tumor cells to evade androgen receptor (AR) pathway inhibitors such as enzalutamide by shedding their dependence on the AR pathway[4,9]. Therefore, understanding the cellular and molecular basis underlying neuroendocrine differentiation (NED) of prostatic tumor cells is of important clinical significance.

The normal prostate gland consists predominantly of cells of the luminal and the basal compartment with a small minority of NE cells that are scattered between the luminal and the basal cell compartment[10]. Several recent studies have attempted to uncover the cell of origin of focal NED and even NEPC. As normal NE cells share many features with malignant NE cells (for example, expressing SYP and CHGA), it has been proposed that NEPC might arise from transformed NE cells[11]. However, genomic studies seem to be supportive of an epithelial origin of NEPC, given that NEPC showed genomic overlap with PCA, such as TMPRSS2-ERG fusion[12,13]. Within the prostatic epithelial cell compartments, both luminal and basal epithelial cells have been shown to be capable of generating prostate cancer and even NEPC. For example, Zou et al.[14] have demonstrated that focal NED, as well as eventual well-differentiated neuroendocrine disease directly arises via transdifferentiation from luminal adenocarcinoma cells. In contrast, Lee et al.[15] have recently reported that basal cells can directly give rise to NE cells during prostatic tumorigenesis without undergoing an intermediate luminal state. In addition, some studies have suggested that NE cells derived from basal cells exhibit a loss of basal features and upregulation of luminal features during NED[16,17]. Overall, there is no consensus on the cellular characteristics during the transition from epithelial tumor cells to neuroendocrine (NE) tumor cells.

Gene expression is a key determinant of cellular phenotypes. Previous population-based RNA sequencing (RNA-seq) method has been performed to compare the transcriptional similarity between prostatic basal and luminal epithelial cells and suggested that metastatic NEPC molecularly resembled stem cell in basal compartment[18,19]. Recent advance in single-cell RNA sequencing (scRNA-seq) technology has greatly empowered us to gain a more comprehensive understanding of the transcriptional signatures of distinct subpopulations of epithelial cells in human and mouse prostate[20–23]. However, a detailed analysis of the cellular states of NED in primary human prostate cancer at single-cell resolution is still lacking. Herein, we apply scRNA-seq technology to determine the cellular identity associated with NED in human prostate cancer. Our datasets reveal that a luminal epithelial state is highly linked with NED of prostate cancer cells. Furthermore, we show by intra-tumoral RNA velocity analysis that the NE cells are directly generated by luminal-like adenocarcinoma cells. Finally, we dissect the transcriptomic landscape underlying NED and validate single-cell derived NED-related gene signatures in bulk RNA samples. Altogether, our results support the epithelial-NE transdifferentiation model regarding the NED in human prostate cancer and offer fresh insights into cellular states and molecular features associated with this process.

## Results

**Single-cell transcriptional profiling of biopsies from 6 CRPC.** Given that focal NED can be more frequently detected in patients with advanced prostate cancer undergoing ADT but not in primary prostatic adenocarcinoma[24–26], we sought to perform scRNA-seq on tumor biopsies from CRPC patients. In this study, we isolated fresh cells from six CRPC patients, four out of whom were found to have low PSA levels (<20 ng/ml; Table 1, Fig. 1A and Supplementary Fig. 1), indicating a higher likelihood of having NED. In these patients, three had received the first-line therapy of the LHRH analog goserelin coupled with the AR inhibitor bicalutamide, two had undergone surgical castration coupled with bicalutamide, while the remaining one was diagnosed as small-cell NEPC at the beginning and treated with chemotherapeutic drug docetaxel. By pathological examination, biopsy tissues from three patients (#2, #5, and #6) displayed cellular morphology resembling small-cell carcinoma and biopsies from patient #1 and #4 presented a classical PCA phenotype (Fig. 1B). However, biopsy from patient #3 was characterized as prostatic intraepithelial neoplasia, which may due to the inaccuracy of the biopsy procedure. The clinical and pathologic features of the biopsy samples are summarized in Table 1.

Then, single-cell suspension from each tissue was subjected to scRNA-seq by a 10x Genomics-based single-tube protocol with exclusive transcript counting through barcoding with unique molecular identifiers[27]. After exclusion of red blood cells as well as cells not passing quality controls, we obtained a total of 21,292 high-quality cells at ~2884 genes detected on average per cell (Supplementary Fig. 2A and supplementary Table 1). Using an unsupervised graph-based clustering strategy, we manually classified different cell clusters into eight major cell types with canonical markers curated from the literature, including epithelial cells, immune cells (T cells, B cells, myeloid cells, and mast cells), stromal cells (fibroblasts and myofibroblasts), and endothelial cells (Supplementary Fig. 2A, B and Supplementary Data 2).

**NE cells present an epithelial phenotype.** Next, in keeping with our aim to characterize NED, we sought to identify NE cells by evaluating the expression levels of 12 well-known NE markers that have been previously characterized as biomarker or driver genes of NEPC, such as *ASCL1*, *CHGA/B*, and *FOXA2*[24,28,29]. Using Seurat scoring strategy, we detected obvious NED in three patients (patient #2, #5, and #6; Fig. 1C), which is in line with the pathological results. Notably, we found that NE^high cell population detected in these three patients all belong to the epithelial cells instead of the non-epithelial cell compartments (Fig. 1C and Supplementary Fig. 2C, D), supporting an epithelial origin of NED. In addition, we noticed that majority of epithelial cells from patient #2 and #5 were scored for a NE phenotype, while only part of epithelial cells from patient #6 have a NE phenotype (Fig. 1C), manifesting different extent of NED among these three patients. Taken together, single-cell analysis showed that three patients likely have NED and suggested an epithelial origin of NED in human prostate cancer.

**NE cells present a malignant luminal-like phenotype.** Having characterizing an epithelial phenotype of NED, we next focused on epithelial compartment by computationally removing all non-epithelial cells and then performing Pearson correlation analysis on these cells. In order to gain more insight into the molecular features of NED in each patient, we then scored each cell according to different lineage/pathway marker genes including epithelial basal/luminal lineage markers[22], AR signature genes[30–32], EMT as well as stem cell genes[33] (Supplementary Table 2). This analysis revealed that most of epithelial cells from patients #2 and #5

**Table 1 Clinical characteristics of the 6 CRPC patients.**

| Patient ID | Age | PSA level at diagnosis of Pca (ng/ml) | The Gleason score at diagnosis of Pca | The TNM stage at diagnosis of Pca | First-line therapy | Second-line therapy | Time from treatment start to CRPC (mo) | Time from CRPC to now (mo) | PSA level at present (ng/ml) | The TNM stage at present |
|---|---|---|---|---|---|---|---|---|---|---|
| Patient #1 | 82 | 8.88 | 4 + 5 = 9 | cT2cN0M0 | Goserelin, bicalutamide | None | 9 | 0.5 | 5.85 | cT4N0M0 |
| Patient #2 | 82 | 56.53 | 4 + 3 = 7 | cT3bN1M0 | Goserelin, bicalutamide | None | 15.9 | 3.2 | 18.57 | cT3bN0M0 |
| Patient #3 | 86 | 55.16 | 4 + 4 = 8 | cT2cN0M0 | Goserelin, bicalutamide | None | 15.9 | 3.7 | 11.76 | cT2cN0M0 |
| Patient #4 | 78 | >149 | 4 + 3 = 7 | cT4N0M0 | Bilateral orchidectomy, bicalutamide | Docetaxel, Abiraterone | 12.5 | 18.6 | 117.95 | cT4N0M1 |
| Patient #5 | 65 | 15.6 | 4 + 4 = 8 | cT3bN1M1a | Bilateral orchidectomy, bicalutamide | None | 27.8 | 2.6 | 0.9 | cT4N1M1 |
| Patient #6 | 70 | 72.72 | Small-cell NEPC | cT4N1M1 | Docetaxel | | 14.8 | 3.7 | 7.18 | cT3bN0M1 |

PSA prostate-specific antigen, Pca prostate cancer, TNM tumor node metastasis, CRPC castration-resistant prostate cancer, NEPC neuroendocrine prostate cancer.

represent an obvious NED phenotype (Fig. 2A, B). In contrast, epithelial cells of patient #6 were divided into two main groups: a small population of NE-like cells and the remaining majority NE⁻ AR$^{high}$ cells, which illustrating clear intra-tumoral heterogeneity regarding NED. Most importantly, by analyzing cellular phenotypes/states, we found that all NE cells prominently exhibited a luminal phenotype rather than basal phenotype (Fig. 2A, B). Of note, AR scores were extremely low in NE cells, which is consistent with previous findings that AR signaling activity is downregulated in NEPC[34]. Previous studies have suggested that the EMT process and stem cell state might play an important role in inducing NED[35]. However, our analysis demonstrated that only NE cells from patient #5 displayed higher EMT and stemness signature scores.

We next interrogated malignant identities of NE cells by performing inferred copy number variation (CNV) analysis on the basis of the average expression of 101 genes in each chromosomal region[36,37]. Inferred CNV analysis supported a malignant identity of NE cells, as evidenced by remarkable CNVs (Fig. 2C, D and Supplementary Fig. 3A). Interestingly, most basal-like epithelial cells lacked CNVs, suggesting that basal epithelial cells were non-malignant or less malignant. In addition, we noticed that the epithelial cells of patient #3 had very few CNVs (Fig. 2C and Supplementary Fig. 3A), which was consistent with its histologically intraepithelial neoplasia characteristic (Fig. 1B).

In addition to correlation analysis, we also visualized cell–cell similarity by UMAP dimension reduction analysis of the 12,861 epithelial cells (Fig. 2E and Supplementary Fig. 3B). This analysis showed that most KRT5$^+$ basal cells from different samples were grouped together. In addition, we found that the NE cells of patient #6 were separated from the remaining epithelial cells in this patient and located closely to the NE cells from patient #2 and #5, demonstrating a certain degree of transcriptional similarity between NE cells from different patients. Taken together, preliminary analyses revealed a malignant feature of NE cells and showed that most NE cells exhibited a luminal-like AR$^{low/-}$ phenotype.

**Intra-tumoral analyses identify different extent of focal NED.** To better understand the extent of NED in each individual tumor, we next investigated intra-tumoral epithelial diversity. Re-clustering epithelial cells from each tumor combined with heatmap analysis showed that epithelial cell sub-clusters from each sample highly expressed luminal cell markers such as *KRT8* and *KRT18*, while the expression of basal, NE, and AR signature genes exhibited apparent intra- and inter-tumor heterogeneity (Fig. 3A, B and Supplementary Data 3). We thus annotated all epithelial clusters into basal, luminal, and NE subtypes, respectively, according to their transcriptional landscapes. For example, in patient #1, we identified a cluster of basal cells (KRT17$^+$; cluster 1 and 7), several clusters of luminal cells (KRT5$^-$ KRT8$^+$, cluster 0, 2, 3, 4, 5, and 6). This analysis also confirmed the NE phenotype of patients #2 and #5 and showed that most clusters in these two patients uniformly expressed NE markers, manifesting a pure NE phenotype. Particularly, immunofluorescence (IF) result illuminated that NE cells in patient #2 expressed both NE and luminal markers (Supplementary Fig. 4). In addition, epithelial cells of patient #6 consisted of a group of NE cells (cluster 4, expressing ASCL1, CHGA, and CHGB), a group of basal cells (cluster 8, expressing KRT5, KRT14, and KRT15) as well as the remaining AR$^{high}$ luminal cells, presenting mixed features of both adeno-carcinoma and NEPC (Fig. 3B). The most interesting observation was from patient #4, a histologically diagnosed adenocarcinoma, in which we found that when compared with other epithelial

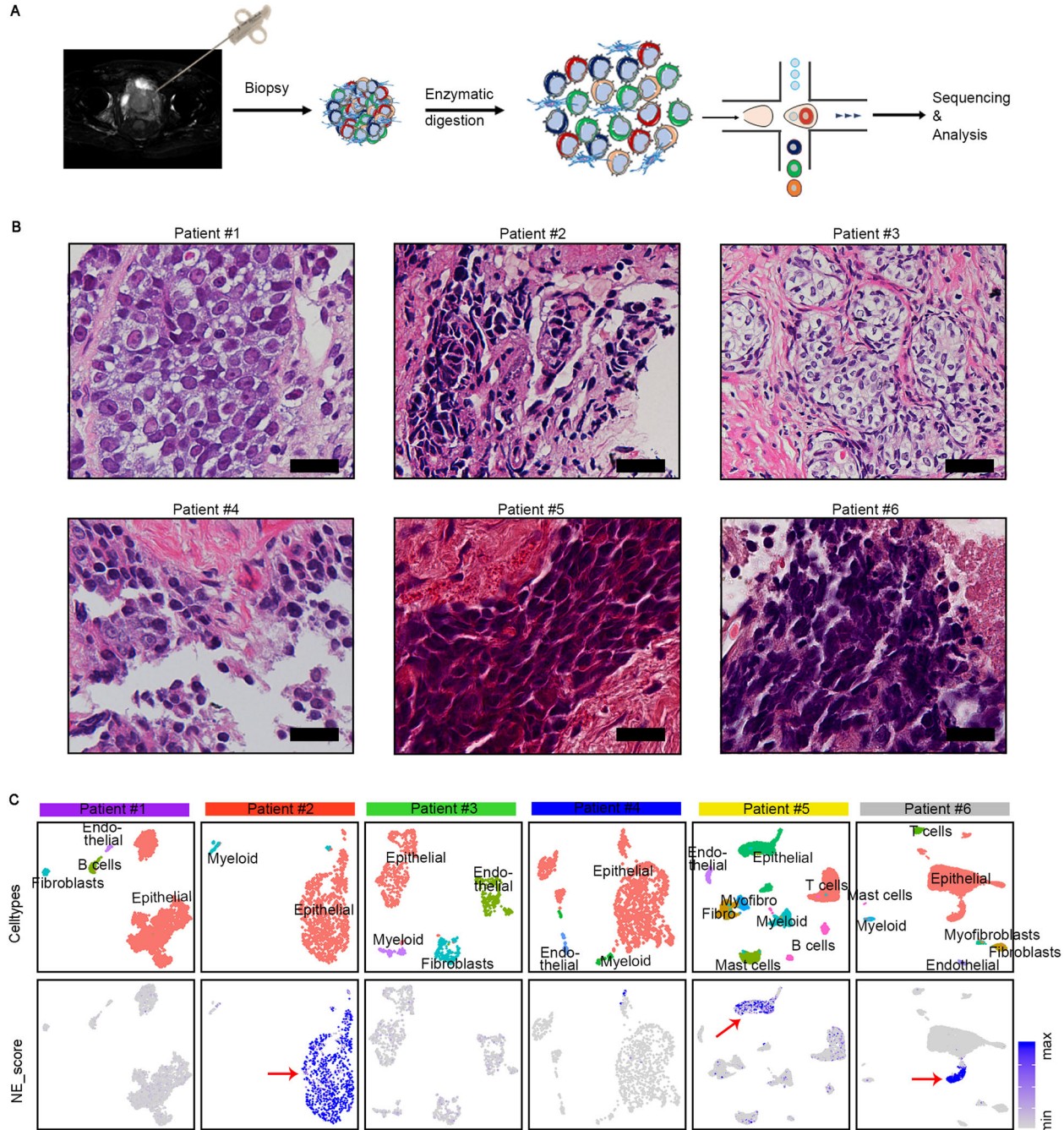

**Fig. 1 Single-cell transcriptomic profiling of 6 CRPC tumors. A** Workflow for single-cell extraction, sequencing, and analysis. **B** Haematoxylin and eosin (H&E) staining for 6 CRPC patients. The scale bars represent 25 μm. **C** UMAP plots of cells from six patients with cells colored based on the cell types (upper row) and NE scores using the well-established NE marker genes (lower row). The minimum score is indicated by light gray and the maximum score is indicated by blue. The red arrows pointed to high NE score cell population.

clusters, cluster 5 preferentially expressed NE markers CHGA and SYP (Fig. 3B), probably representing a population of early NE precursors. These observations were further validated by IHC assays for lineage markers in sections from five samples (Fig. 3C). For instance, we detected a minority of scattered SYP+ NE cells in section from patient #4, which may correspond to the cells of cluster 4 revealed by single-cell analysis. In addition, IHC analyses of patient #5 samples also showed an overall good concordance with the single-cell transcriptional profiles that SOX2 was intensively expressed, while another NE marker SYP was almost undetectable (Fig. 3C). Thus, intra-tumoral analysis

confirmed NED in three patients (patient #2, #5, and #6) and enabled us to detect NE cells in a PCA (patient #4).

**Epithelial cellular relationships in patient #4.** We next paid specific attention to patient #4, given that the NE subpopulation detected in this PCA may represent an early state of transdifferentiation from epithelial toward NE fate. Epithelial cells in patient #4 were partitioned into four main subtypes: basal cells (cluster 6, expressing KRT5 and TP63), urothelial-like cells (cluster 4, expressing UPK1A and GATA3), NE cells (cluster 5, expressing SYP and EZH2), and luminal cells with a

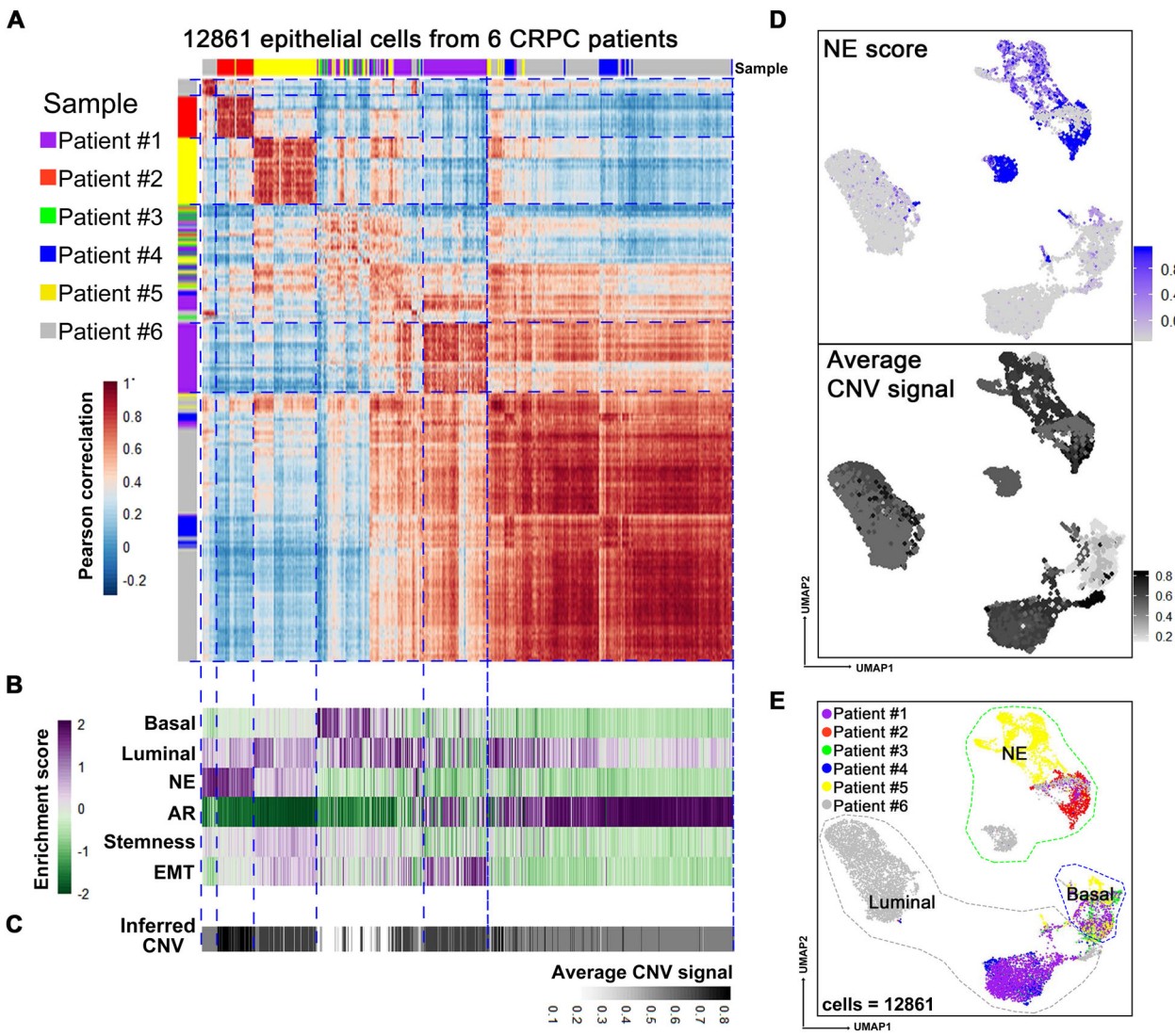

**Fig. 2 NE cells present an epithelial phenotype. A** Pairwise correlations between the expression profiles of 12,861 epithelial cells (rows, column) from 6 CRPC samples (color bar). Correlations were calculated across 63 lineage-specific genes (Supplementary Table 2). **B** Enrichment scores for gene lists including basal, luminal, NE, AR, stemness, and EMT pathway associated genes. Cells were ordered as in (**A**). Green indicates a low score and purple indicates a high score. **C** Average inferred CNV signals of corresponding cells in (**A**). Black indicates the high CNV signal (Supplementary Fig. 3). **D** UMAP visualization of all 12,861 epithelial cells for the 6 patients with cells colored by the gradient of NE score (top) and average CNV signal (bottom). The minimum score is indicated by light gray and the maximum score is indicated by blue (top) or black (bottom). **E** UMAP visualization of all 12,861 epithelial cells from 6 patients with color-coded for the sample origin which kept concordant with (**A**).

KRT5⁻UPK1A⁻SYP⁻KRT8⁺ feature (clusters 0–3; Fig. 4A, B). UMAP visualization suggested that NE cells were transcriptionally closer to luminal cells than basal or urothelial-like cells. IF analysis of SYP and KRT8 further validated a luminal phenotype of SYP-expressing cells (Fig. 4C). Interestingly, the early NED cells and luminal cells shared almost the same CNV pattern, indicating that they had a common clonal origin (Fig. 4A and Supplementary Fig. 5A). In contrast, basal cells in this sample displayed very few CNVs. Thus, the separation of different epithelial subtypes may reflect their marked genomic differences.

A closer relationship between NE cells and luminal-like malignant cells was further supported by visualization using Partition-based approximate graph abstraction (PAGA)[38] (Fig. 4D). To deepen our understanding of the dynamics of epithelial to NE transition, we next applied RNA velocity analysis that predicts the future state of an individual cell by leveraging the fact that newly transcribed, unspliced pre-mRNAs and mature, spliced mRNAs can be distinguished in common single-cell

RNA-seq protocols[39]. Notably, unlike many other existing computational methods[40], RNA velocity analysis does not rely upon a priori knowledge to set up the starting cell for inferring the trajectory and thus enable us to more unbiasedly and accurately predict the cellular differentiation trajectory. Given the heterogeneous epithelial composition, we utilized scVelo, a likelihood-based dynamical model that has recently be introduced to solves the full gene-wise transcriptional dynamics[41]. This analysis clearly showed positive velocity from luminal malignant cells (cluster 3) toward early NED cells (cluster 5; Fig. 4E). In contrast, KRT5⁺ basal and UPK1A⁺ urothelial-like cells were clustered far from NED cells and did not show a tendency to progress into SYP⁺ cells. Therefore, this finding suggested that luminal-like malignant cells may serve as the direct progenitor cells responsible for early NED in this patient we analyzed here.

**TMA analysis confirms the prevalence of luminal-like NED phenotype.** We next validated the generality of this observation

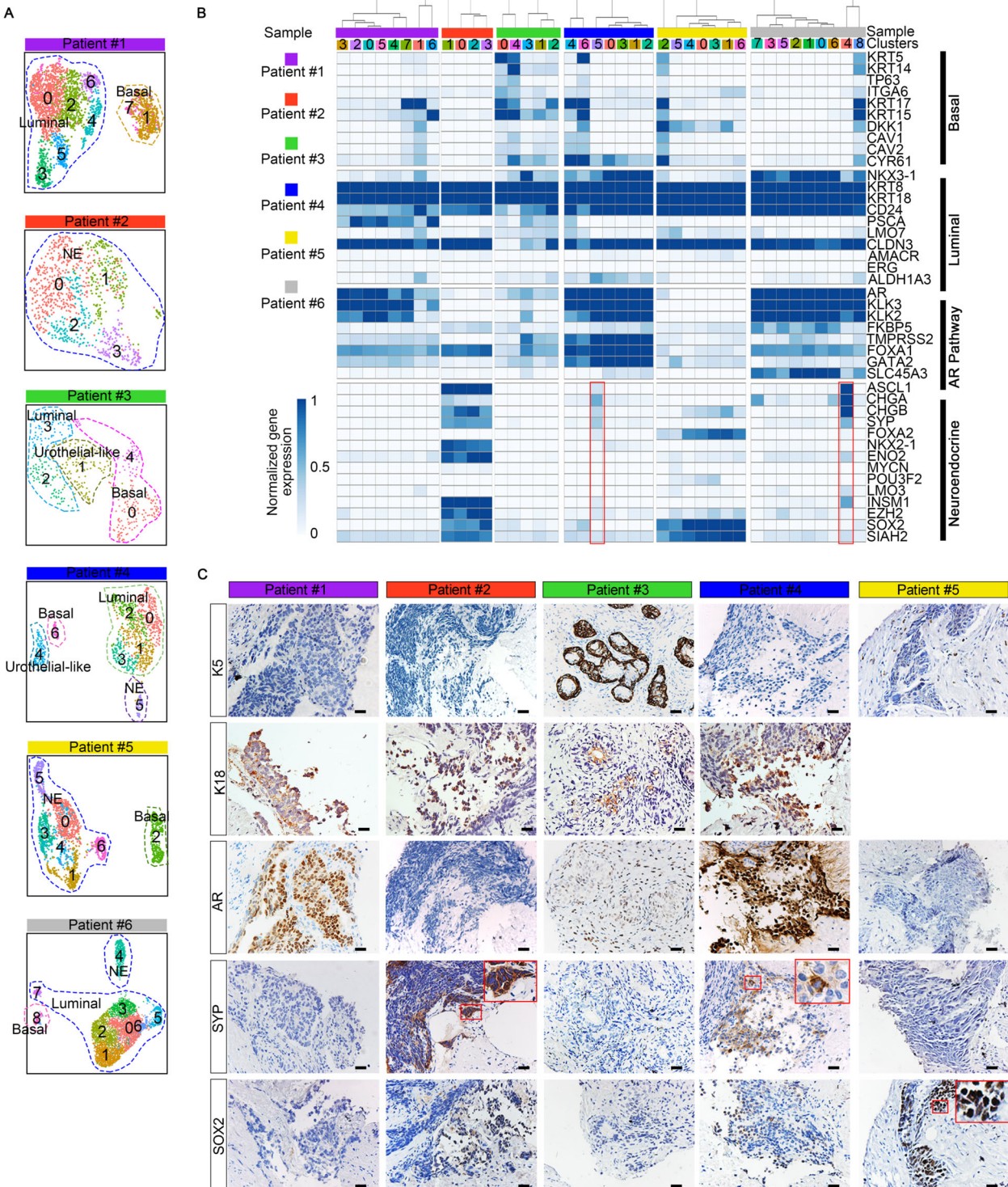

**Fig. 3 Intratumor heterogeneity analyses reveal different extents of NE differentiation. A** UMAP visualization of epithelial cell sub-clusters from each sample. **B** Heatmap depicting prostate lineage marker genes and AR pathway gene expression levels in epithelial cell sub-clusters from each sample. Those highlighted in red frame showed cluster 5 in patient #4 and cluster 4 in patient #6 was NE sub-clusters. **C** Immunohistochemistry (IHC) staining for K5, K18, AR, SYP, and SOX2 in sections from five samples. Scale bars represent 50 μm.

in a large population using clinical PC TMAs, which contained 297 cancer tissues (280 PCA, 10 CRPC, and 7 NEPC) (Supplementary Data 4). We carried out triple IF staining for K18, K5, and SYP to evaluate the basal/luminal phenotypes of NE cells (Fig. 4F). Consequently, we detected SYP-positive cells in 102 tumors, of which 81% were K18+K5−SYP+, and 5% exhibited both K18+K5−SYP+ and K18−K5−SYP+ characteristics

(Fig. 4G). Notably, no K18−K5+SYP+ cells were found in any of the 297 cancer tissues. This analysis therefore verified that NED precursors in human prostate cancer had a prevalent luminal phenotype. Of interest, a substantial number of the SYP-expressing tumor specimens came from patients who had not received any therapy (96/102), demonstrating that NED in fact occurred much earlier than the development of castration

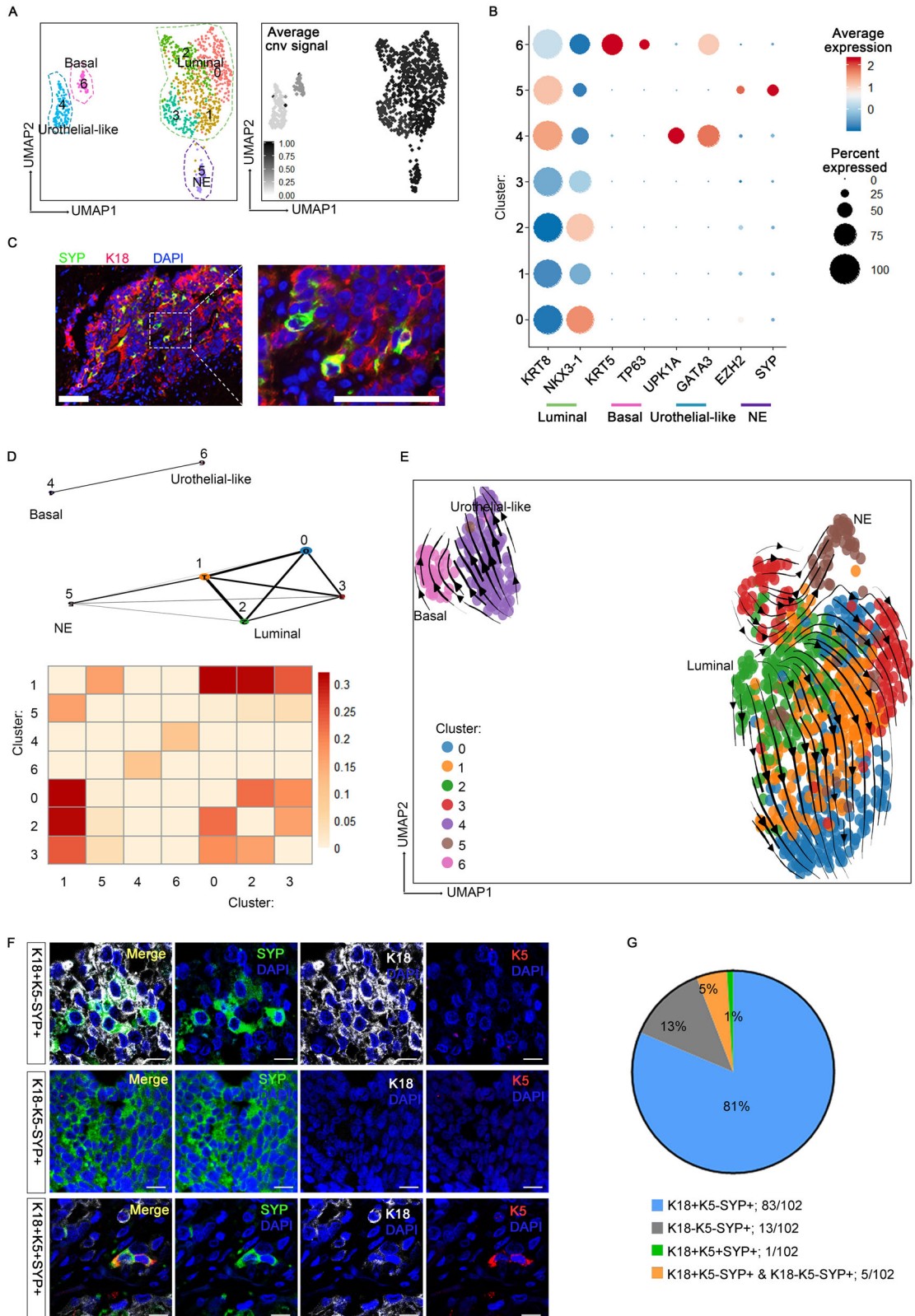

resistance, which is in line with previous findings that neuroendocrine differentiation is present in 10–100% of localized PCAs and increases with disease progression[42,43]. All together, TMA analysis generally supported the single-cell results, showing that the most of NED cases in human prostate cancer exhibited a luminal-like phenotype.

**Epithelial cellular relationships in patient #6.** Similar to patient #4, epithelial cells of patient #6 also showed intra-tumoral NED heterogeneity, which was composed of a small population of NE cells (cluster 4), a small population of basal cells (cluster 8), and the vast majority of luminal epithelial cells (Fig. 5A, B). Interestingly, like patient #4, basal epithelial cells in patient #6

**Fig. 4 Epithelial cellular relationships in patient #4. A** UMAP visualization of epithelial cells from patient #4 with color-coded for the corresponding sub-cluster (left) and the average inferred CNVs signals (right; gray to black). **B** Dot plots of the expression level of NE, urothelial-like, basal and luminal lineage markers across the populations shown in (**A**) (Source data are provided as Supplementary Data 1). **C** Immunofluorescence (IF) co-staining for K18 (red) and SYP (green) in sections for patient #4. Scale bar represents 100 µm. **D** The PAGA graph and connectivity scores of the populations shown in (**A**). **E** Velocities of epithelial cells from patient #4 are visualized as streamlines in a UMAP-based embedding, in which color-coded for the corresponding populations shown in (**A**). **F** Representative confocal fluorescence microscopy of triple co-staining of SYP (green), K18 (gray), and K5 (red) in PC TMA sections. The SYP + NE cells have three subtypes: K18+K5−SYP+, K18−K5−SYP+, and K18+K5+SYP+. Scale bars represent 25 µm. **G** Pie chart of statistics for PC TMA co-staining results showing that the major part of prostate cancers contain NE cells with exclusive luminal phenotype (K18+SYP+,83/102).

epithelial cells also displayed relatively fewer CNVs compared with luminal compartment as well as NE cells (Fig. 5A and Supplementary Fig. 5B), indicating that basal epithelial cells were less likely to be the direct progenitors of NE cells. The cellular relationship was further indicated by PAGA (Fig. 5C), showing that NE cells in this sample still connected with luminal-like tumor cells. We next inferred cellular dynamics using RNA velocity, which predicated similar cellular processes that NED in this sample was exclusively branched from luminal cells (Fig. 5D). We further sought to identify genes that display pronounced dynamic expression patterns linked to the transition state toward a NE fate (Supplementary Data 5). As expected, signatures of AR signaling such as KLK2 and KLK3 were notably downregulated along with the emergence of NE phenotype (Fig. 5E). We then paid attention to genes that were positively correlated with NED. Within the top-ranked likelihood genes, we found ASCL1, a key transcription factor for neuronal differentiation[44], which has also been associated with NED in prostate cancer[45] (Fig. 5E). In addition, this analysis also illustrated many unknown genes that might serve as the potential drivers or biomarkers of the NED transdifferentiation, for example, VGF, SCGN, and PAPPA2, the roles of which in NEPC have not been reported. Altogether, deeper analyses of epithelial cell relationships in this sample also suggested that malignant cells with a luminal phenotype fuels the development of NE cells.

**Identifying NED-associated gene meta-programs.** We next sought to understand the underlying molecular features associated with NED. For this purpose, we applied non-negative matrix factorization (NMF) to define underlying transcriptional programs specific to the epithelial cells from each tumor[46,47] (Fig. 6A and Supplementary Data 6). To relate these meta-programs to cell phenotypes, we scored these ordered cells according to basal, luminal, NE, EMT, AR, and cell cycle marker genes (Fig. 6B). This analysis revealed three meta-programs highly associated with NED (P1, P2, and P4). For example, meta-program P1 was characterized by neuroendocrine markers such as *CHGB* and *CHGA* and meta-program P2 contained NE-related transcriptional factor (TF) *EZH2* and *DLX5*, a homeobox transcription-factor gene. *DLX5* has been recently reported to mark delaminating neural crest cells during development[48]. Of note, neural crest cells can differentiate into numerous derivatives including neuroendocrine cells[49,50], implying a potential role of this gene in participating NED of prostate cancer cells. Moreover, we identified a cell cycle-related meta-program (P3) that was obviously upregulated in NE cells of patient #2 and #5), likely reflecting well-differentiated NE state of these two tumors. More interestingly, meta-program P2 was specifically associated with patient #2, while meta-program P4 was preferentially expressed in patient #5, suggesting two kinds of NED features.

We next asked whether the NE-related gene meta-programs derived from single-cell data could robustly reflect the NED in bulk expression profiles. Thus, we used three bulk-transcriptomic datasets[7,30,51], which included both CRPC and NEPC patients. We first performed correlation analysis between the expression of

all genes from three meta-programs (P1, P2, and P4) and the NE score defined by the average expression of well-established NE markers to screen out genes that were most relevant to NED. This analysis identified 121 genes highly correlated with the NE score (Pearson $R \geq 0.3$; Fig. 6C and Supplementary Data 7). Consistently, we found that by plotting their expression in the five groups of samples that was defined by the expression patterns of NE and AR activity genes[30], most genes displayed evidently higher expression in the AR−NE+ group than in NE− groups (Fig. 6D). Thus, NED-associated gene signatures derived from single-cell data can provide reliable clues for distinguishing human NEPC and searching for undescribed drivers involved in NED.

**Identifying NED-associated transcription-factor regulatory network.** The above NMF analysis revealed that two well-differentiated NEPC displayed distinct NED signatures. To explore the underlying molecular mechanisms driving the distinct NE differentiation phenotypes, we next used single-cell regulatory network inference and clustering (SCENIC) to identify the co-expressed transcription factors and their putative target genes, as an indicator of transcription-factor regulatory activity[52]. SCENIC analysis showed that NED from different patients could upregulate the expression of different transcription-factor networks (Fig. 7A). For instance, DLX6 and ASCL1 regulons were highly active in NE cell of patient #2, whereas expression of FOXA2 and SOX21 network was restricted in NE cells of patient #5. In line with reports that SOX2 is essential for NED in prostate cancer, we found that SOX2 regulon was upregulated across almost all NE cells from patient #2 and #5 (Fig. 7A). Thus, single-cell regulatory network analysis provided an explanation for the divergence of NED from our patient cohort. In addition to many well-established NE-related TFs identified in this analysis (such as *SOX2* and *ASCL1*), it also predicted many neuronal differentiation-related TFs that might also be involved in NED. For instance, expression of *LHX2* has previously been shown to confer neuronal competency for activity-dependent dendritic development of cortical neurons[53], but its role in NED of prostate cancer remains undetermined and need future studies to clarify their specific roles.

Next, we analyzed TF regulons of epithelial cells from two patients with intra-tumoral NED heterogeneity. Analysis of patient #4 revealed that NE subpopulation specifically upregulated transcriptional activities of NKX2-2, HES6, FOXA2, and ASCL1, all of which have been previously reported to be essential for a variety of neural cell types' differentiation (Fig. 7B, C). The intra-tumoral heterogeneity in terms of TF activity was also observed in patient #6, showing that NE subpopulation has obviously higher TF activities of SOX2 and FOXA2 (Fig. 7D, E). In addition, NE subpopulation strongly upregulated activities of UNCX and CELF5 regulatory networks, which have been both reported to involve in maintaining neural cell survival or promoting some neuron diseases[54,55]. Overall, TF network analysis revealed both known and unknown NED-associated

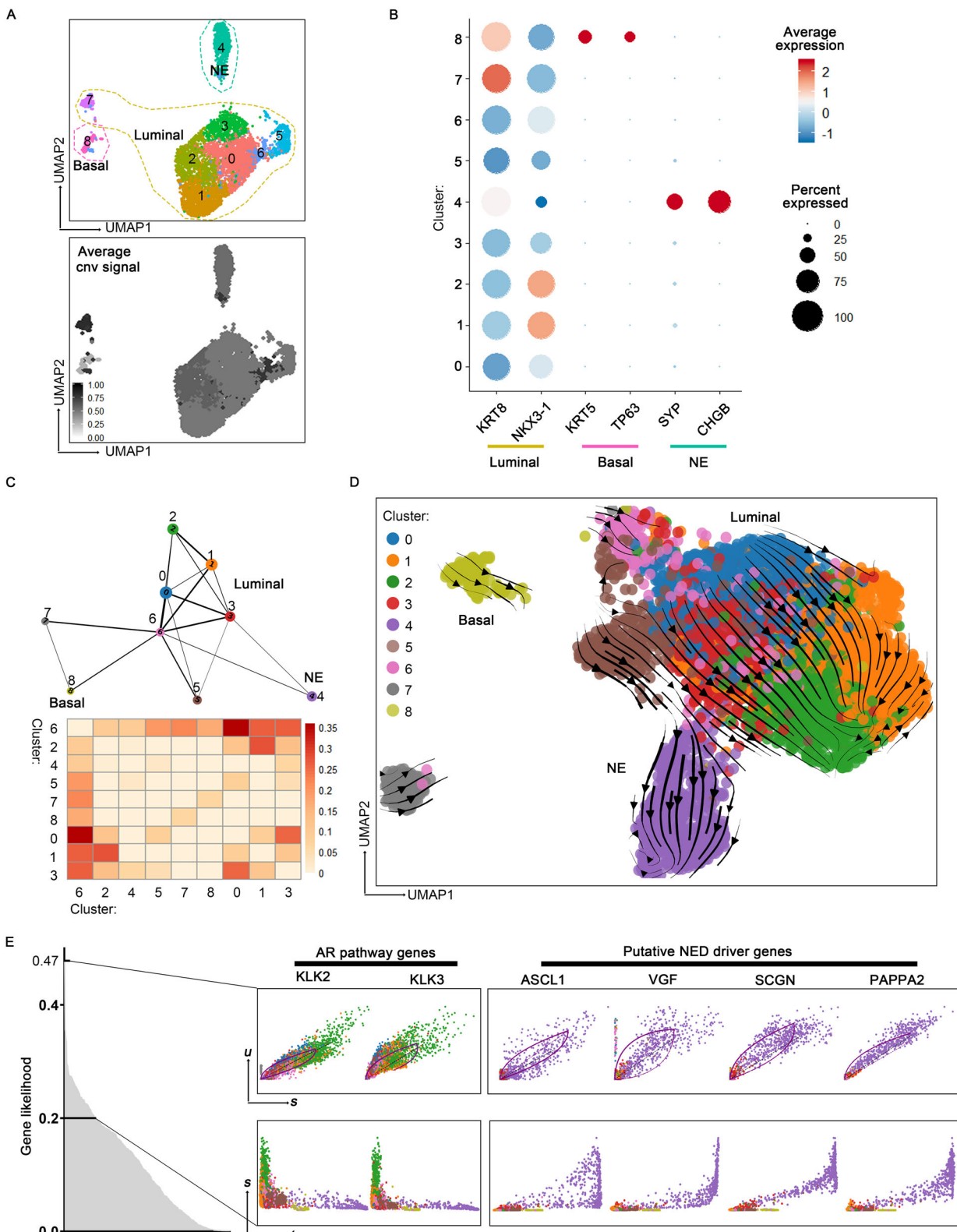

**Fig. 5 Epithelial cellular relationships in patient #6. A** UMAP visualization of epithelial cells from Patient #6 with color-coded for the corresponding sub-cluster (top) and the average inferred CNV signal (bottom; gray to black). **B** Dot plots of the expression level of NE, basal, and luminal lineage markers across the populations shown in (**A**) (Source data are provided as Supplementary Data 1). **C** The PAGA graph and connectivity scores of the populations shown in (**A**). **D** Velocities of epithelial cells from patient #6 are visualized as streamlines in a UMAP-based embedding, in which color-coded for the corresponding Seurat cluster in (**A**). **E** Phase portraits (upper row) and expression dynamics along latent time (lower row) for specific genes selected from top-ranked likelihood gene set (gene likelihood >0.2).

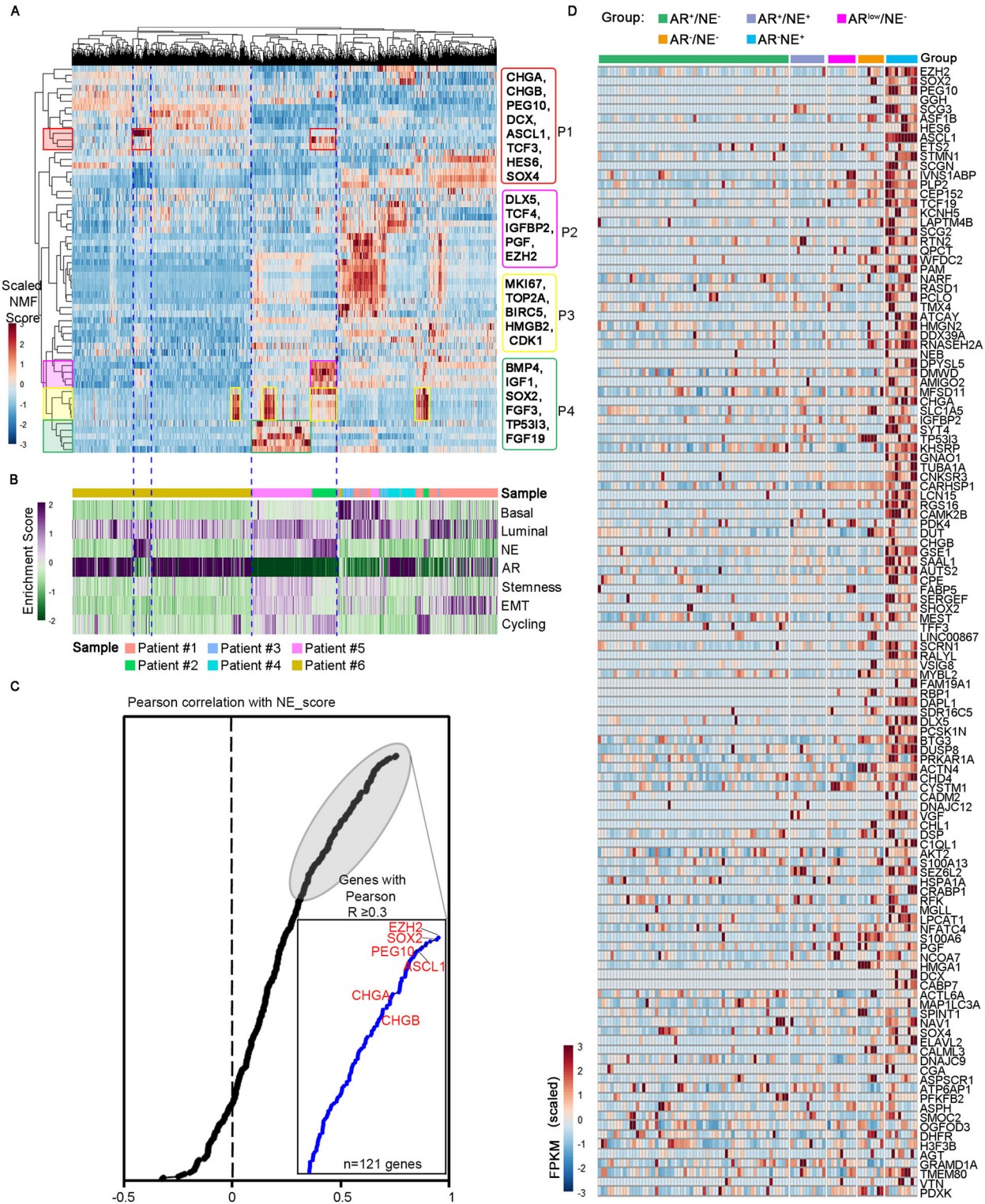

**Fig. 6 Intra-tumoral meta-programs underlying NED. A** Heatmap showing scores of 12861 epithelial cells (column, from 6 CRPC patients) for each of 60 programs (rows) derived from NMF analysis of individual samples. Cells and programs are hierarchically clustered, and 3 NE-related meta-programs (P1, P2, and P4) and a cell cycle-related meta-program (P3) are highlighted. **B** Enrichment scores of prostate lineages: basal, luminal, NE marker genes and AR, stemness, EMT, and cell cycle pathway genes in cells ordered as in (**A**), with the color-coding for the corresponding CRPC sample. **C** Pearson correlation between the expression of genes of P1, P2, and P4 and the NE score, as measured by the average expression of 14 known NE markers. Three previously published bulk RNA-seq datasets were used in this analysis, as described in the "Methods" section. Highlighted in red are some known NED genes (Source data are provided as Supplementary Data 1). **D** Heatmap depicting strong expression of 121 genes (Pearson $R \geq 0.3$, as measured by Pearson correlation analysis shown in (**C**) in AR−NE+ group of Morrissey dataset. Total samples are divided into five groups as previously suggested in ref. [40].

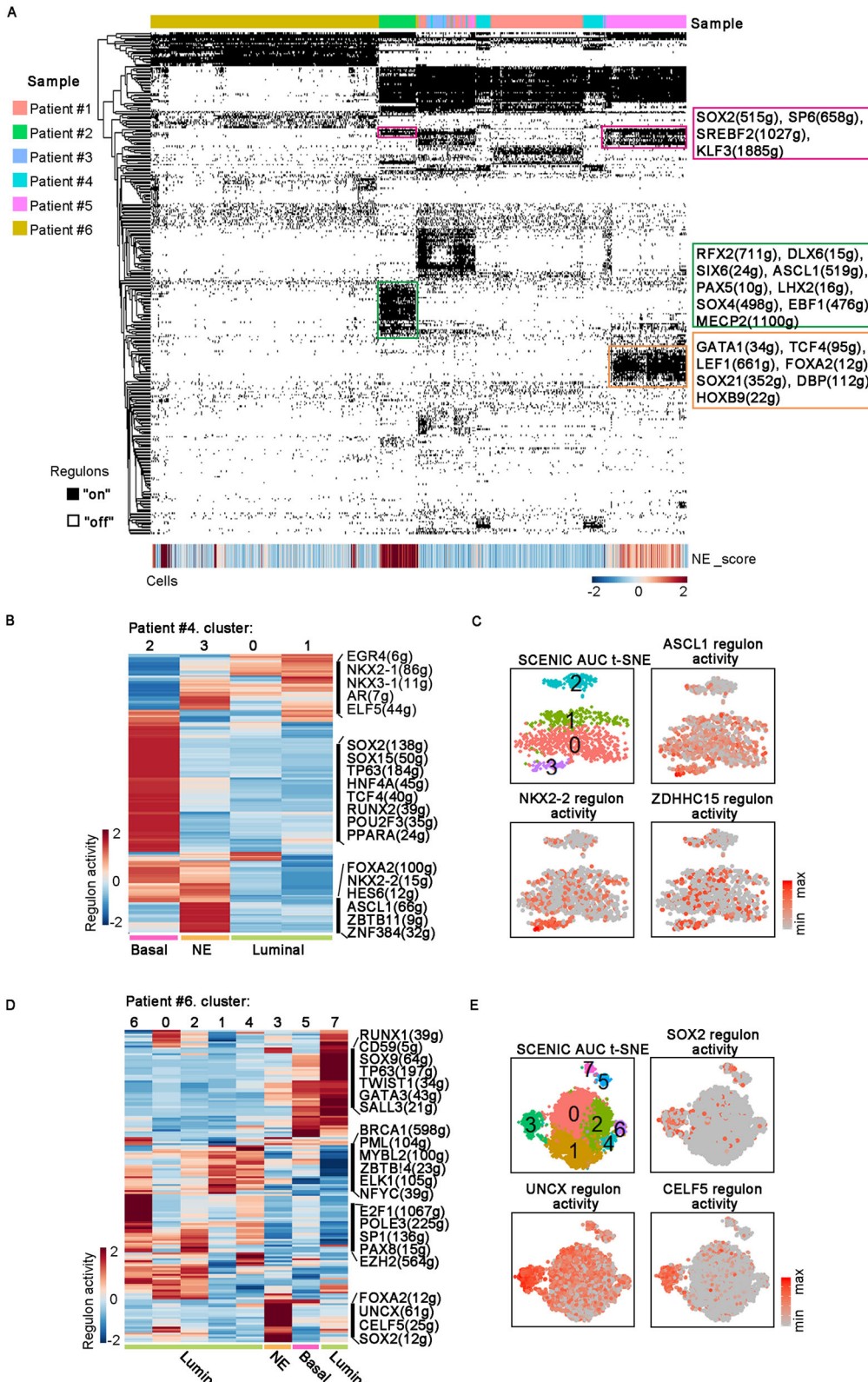

**Fig. 7 Transcription-factor regulatory networks underlying NED. A** Heatmap of SCENIC binary regulon activities (row) and NE scores (row) of 12,861 epithelial cells (column). Three TF regulatory networks with high activities in NE cells were highlighted. **B** Heatmap of the mean regulon activities (row) that differentially expressed on epithelial clusters (column) of patient #4. **C** t-SNE on the SCENIC regulon activity matrix and the representative regulon activities on epithelial cells from patient #4. Cells are colored by the corresponding cluster and gradient of regulon activity (gray to red). **D** Heatmap of the mean regulon activities (row) that differentially expressed on epithelial clusters (column) of patient #6. **E** t-SNE on the SCENIC regulon activity matrix and the representative regulon activities on epithelial cells from patient #6. Cells are colored by the corresponding cluster and gradient of regulon activity (gray to red).

TFs and offered more insight into both intertumoral and intra-tumoral heterogeneity regarding NED.

## Discussion

In this study, we generated 21,292 single-cell transcriptomes from 6 CRPC patients with a focus on the cellular phenotypes associated with NED. We detected NED in four tumors, in which all of the NE cells exhibited a luminal rather than basal epithelial phenotype. It is important to note that in two tumors that contain both NE cells and non-NE epithelial cells (patient #4 and #6), there is clear cell fate transition tendency from luminal-like adenocarcinoma cells toward NE cells (Figs. 4E, 5D). Thus, our finding has identified the transdifferentiation process that has been proposed for a long time in explaining NED in prostate cancer. Although previous genomic analyses have suggested that NEPC are clonally derived from PCA that usually present luminal-like phenotype[7,12,13], this is the first study to our knowledge that has shown the cellular diversity in human CPRC as well as the cellular phenotypes associated with NED at single-cell resolution.

Our current study is limited regarding the total number of samples that contain NED for analyses. To unbiasedly evaluate the cellular phenotypes associated with NED, we next performed triple IF staining against KRT5, KRT8, and SYP on our large cohort of PC TMAs. We found that the luminal-like malignant phenotype of NE cells (K5$^-$K18$^+$SYP$^+$) is mainly detected in adenocarcinomas (Fig. 4G). Therefore, this results further confirmed a closer relationship between NED and a luminal state rather than basal state in human prostate cancer. It should be noted that our results do not exclude the probability that basal cells serve as cells of origin of NEPC. According to in vivo cell lineage tracing studies, both basal and luminal cells are capable of initiating prostate tumorigenesis[56]. In particular, prostate cancer originated from human basal cells gradually loss basal features and upregulation of luminal hallmarks[16,57]. Based on these findings and our current results, we propose a model that PCA can be initiated from both basal and luminal cells, while focal and eventual NEPC is more likely to be made by NE precursors with luminal phenotype (Fig. 8). We also consider the possibility that a direct basal-NE transdifferentiation may happen. If NE cells are directly transdifferentiated from basal cells, we would expect to see hybrid cells with both basal and NE phenotypes more frequently. However, our analysis in PC TMAs reveals that only about 1% of patients (1/102) contain SYP$^+$ cells that express both K8 and K5 in adenocarcinoma tissues argue strong for the notion that such direct basal-NE transdifferentiation is likely rare in human prostate cancer, but rather luminal-NE transdifferentiation is fundamentally responsible for phenotypic transition from acinar adenocarcinomas toward NEPC. Interestingly, a recent cell lineage tracing study using TRAMP mouse models (p63-CreERT2;Rosa-LoxP-STOP-LoxP-tdRFP;TRAMP and K8-CreERT2;Rosa-LoxP-STOP-LoxP-tdRFP;TRAMP) has demonstrated that NEPC is directly originated from basal progenitor cells but not luminal cells or pre-existing KRT8$^+$ adenocarcinoma cells[15]. This observation is different from the results obtained from the double *p53* and *Pten* knockout-induced PCA mouse model in which all NEPC cells were transdifferentiated from NKX3.1-expressing luminal cells. According to our results, we are inclined to think that a transformed basal cell would first differentiate to a luminal-like tumor cell and then execute NED process. Nevertheless, future single-cell studies of serial tumor samples from individuals will be needed in principle to map the cellular dynamic involved in NED process as much as possible.

Our next aim is to explore the signature genes driving NE transdifferentiation. By performing NMF analysis, we further identified three gene meta-programs consisting of many genes highly correlated with NED (Fig. 6A). The bulk datasets analysis has validated the robustness of this result, showing that most genes are evidently expressed in patients with NED (Fig. 6D). Interestingly, we found that two well-differentiated NEPCs (patient #2 and #5) seem to have distinct NED programs. SCENIC analysis highlighted that the heterogenous NED might be determined by distinct TF networks. Nevertheless, the exact role of many identified genes in prostate cancer, especially NEPC, is unknown and needs further comprehensive investigation.

In summary, our single-cell study has disentangled both intra- and intertumoral heterogeneity regarding NED in human prostate cancer and characterized both cellular phenotypes and

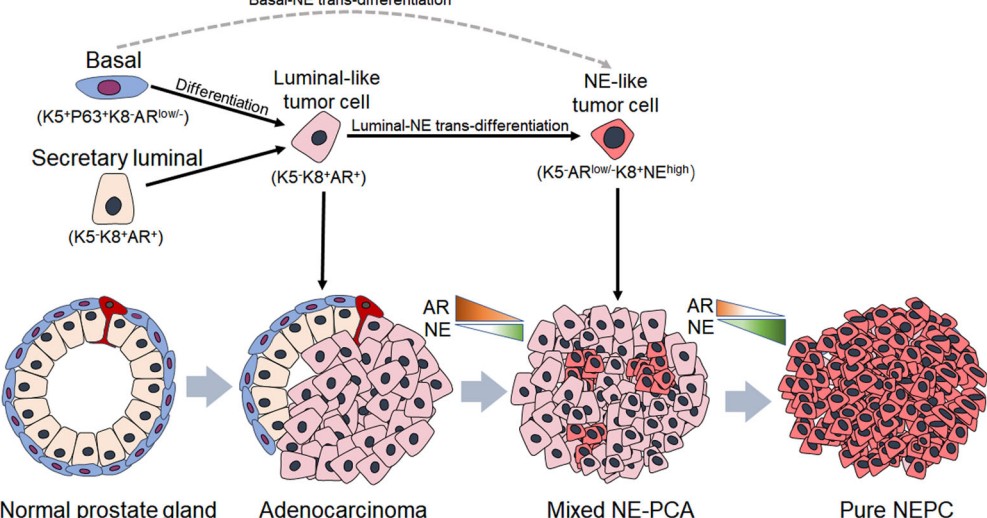

**Fig. 8 Cellular relationship and disease progression model of NEPC.** Schematic illustration of tumor evolution toward the neuroendocrine phenotype, in which dotted arrows indicate the potential relationship between cell lineages and the solid arrows indicate that NEPC is directly originated from AR-dependent tumor cells. In this model, we suppose that the NE precursor, AR-independent tumor cell, directly transdifferentiates from the luminal-like tumor cell, and that is the precursor, which will next evolve in forming the focal NEPC and finally progress to small-cell (pure) NEPC. The extent of AR and NE signature scores varies over the spectrum of adenocarcinoma to neuroendocrine transdifferentiation (orange indicates a high level of AR signal and green indicates a high level of NE signal).

molecular features linked with the luminal to NE transdifferentiation. Understanding the progressive trajectory of NED will benefit the development of early diagnosis and even therapeutic treatments for human NEPC.

## Methods

**Patient selection**. With a focus on neuroendocrine prostate cancer, the participating patients were required to meet the following requirements: (1) the patients must have developed resistance to castration therapy; (2) CT imaging showed an apparent prostate tumor (Supplementary Fig. 1A). In addition, we preferentially selected patients whose circulating PSA level was lower than 20 ng/ml. The patient information is described in detail in Table 1. The present study was approved by the Institutional Ethics Review Board of Ren Ji Hospital, Shanghai Jiao Tong University School of Medicine, and written informed consent was obtained from every patient.

**Isolation of single cells**. Prostate biopsies were transported to the research laboratory on ice in DMEM/F12 (Gibco, 11320033) with 3% FBS (Gibco, 10099-141) within 30 min of collection. Each specimen was equally separated into two fragments. One fragment was processed for histopathological assessment, and the remainder of the provided tissues was processed for scRNA-seq. In brief, fresh tumor samples were minced and place in a 1.5 ml Eppendorf tube, where they were enzymatically digested with collagenase IV (Gibco) and DNase I (Sigma) for 1 h at 37 °C with agitation. After digestion, samples were sieved through a 70-μm cell strainer, washed with 1% BSA and 2 mM EDTA in PBS, and centrifuged for 5 min at $350 \times g$. Single-cell suspensions were subjected to Lympholyte-H separation (Cedarlane, CL5020) to remove RBCs and debris according to the manufacturer's specifications. Pelleted cells were then resuspended in DMEM/F12 with 3% BSA and were assessed for viability and size using a Countess instrument (Thermo).

**Single-cell library preparation and sequencing**. A total of 5000 cells per sample were targeted for capture. Then, the cell suspension of each sample was run in the Chromium Controller with appropriate reagents to generate single-cell gel bead-in-emulsions (GEMs) for sample and cell barcoding. The libraries were then pooled and sequenced on a NovaSeq 6000 (Illumina) at a depth of ~400 M reads per sample.

**Single-cell data preprocessing and quality control (QC)**. Raw sequencing data were converted to FASTQ files with Illumina bcl2fastq, version 2.19.1, and data were aligned to the human genome reference sequence (GRCH38). The CellRanger (10X Genomics, 2.1.1 version) analysis pipeline was used for sample demultiplexing, barcode processing, and single-cell 3′ gene counting to generate a digital gene-cell matrix from these data. Of note, Cell Ranger filters any barcode that contains <10% of the 99th percentile of total UMI counts per barcode, as these are considered to be associated with low-quality cell barcodes. This processing resulted in an average of 160,233 reads per cell, and an average of 2884 genes were detected per cell (Supplementary Table 1). The gene expression matrix was then processed and analyzed by Seurat (version 3.0) and an R toolkit (https://github.com/satijalab/seurat), using the software R (version 3.6.0). We performed Seurat-based filtering of cells based on the number of detected genes per cell (500–7000) and the percentage of mitochondrial genes expressed (<10%). The mitochondrial genes and ribosomal genes were also removed from the gene expression matrix. Following quality control, 21,292 high-quality cells were retained with an average of 2419 genes were detected per cell (Supplementary Table 1). Each single-cell dataset was then processed by *SCTransform* from the *Seurat* package, which contained the function of normalization, regression, and identification of variable genes.

**UMAP visualization and cell-type annotation**. We used UMAP[58] to visualize the clusters of cells that passed quality control for each sample. Clusters were associated with cell types based on the expression of differential expression of well-established marker genes for each cell type: T cells (*CD2*, *CD3D*, *CD3E*, and *CD3G*), B cells (*CD79A*, *CD79B*, *CD19*, and *MS4A1*), myeloid cells (*CD14*, *CD68*, *AIF1*, and *CSF1R*), mast cells (*MS4A2*, *ENPP3*, *PCER1A*, and *KIT*), fibroblasts (*DCN*, *TNFAIP6*, *APOD*, and *FBLN1*), myofibroblasts (*MYH11*, *GJA4*, *RGS5*, and *MT1A*), endothelial cells (*ENG*, *CLDN5*, *VWF*, and *CDH5*) and epithelial cells (*EPCAM*, *KRT8*, *KRT5*, and *CDH1*[22,37,59–62].

**Defining cell scores**. We used Seurat AddModuleScore function to evaluate the degrees to which individual cells express a certain pre-defined gene set as described previously[37,63]. For example, the NE gene set included *ASCL1*[28], *FOXA2*[64], *NKX2-1*[30], *MYCN*[16], *POU3F2*[65], *INSM1*[66], *SIAH2*[29], NCAM1, CHGA/B, SYP, and ENO2[67] (Fig. 1C). Using the same approach, we defined scores to estimate the activities of prostate cell lineages/pathway corresponding to basal, luminal, NE, AR pathway, EMT state, and cell stemness from previous literatures[4,22,29–33,64,66,68,69] (Figs. 2B, 6B). The detailed gene list can be found in Supplementary Table 2.

**Inferred CNV analysis from scRNA-seq**. Large-scale CNVs inferred from single-cell gene expression profiles using a previously described approach (https://github.com/broadinstitute/inferCNV/wiki)[36,37]. To determine the distinct chromosomal gene expression pattern of epithelial cells in comparison to putative noncarcinoma cells, we set normal prostate epithelial cells from a dataset which contains 78,286 prostate epithelial cells captured by Henry et al. from three health men[22] as the "reference" cells. In addition, those genes expressed in fewer than 200 cells were removed from the count matrix. Average expression was calculated using the log-transformed data (log2[1 + UMI]), and absolute values of fold change were bound by 3. All genes were sorted by their chromosome number and start position. The chromosomal expression patterns were estimated from the moving averages of 101 genes to determine the window size, and they were adjusted as central values across genes. Finally, the average CNV signal was estimated by averaging the CNV modification for 22 autosomes.

**Multiple datasets integration and Batch correcting**. For merging multiple datasets, we applied Harmony integration[70], which has been shown to reduce technical batch effects while preserving biological variation for multiple batch integration. RunHarmony returns a Seurat object, updated with the corrected Harmony coordinates. The manifold was subjected to re-clustering use the corrected Harmony embeddings rather than principal components (PCs), set reduction = 'harmony', with parameters of Seurat analysis.

**Differential gene expression analysis**. DEGs in a given cell type compared with all other cell types were determined with the FindAllMarkers function from the Seurat package (one-tailed Wilcoxon rank-sum test, *P* values adjusted for multiple testing using the Bonferroni correction). For computing DEGs, all genes were probed provided they were expressed in at least 25% of cells in either of the two populations compared and the expression difference on a natural log scale was at least 0.25.

**RNA velocity**. RNA velocities were predicted using velocyto in R program (http://velocyto.org, version 0.6)[39,41]. Briefly, spliced/unspliced reads were annotated by velocyto.py with CellRanger (version 2.2.0), generating BAM files and an accompanying GTF; then, they were saved in .loom files. The .loom files were then loaded to R (version 3.6.0) using the read.loom.matrices function, and they generated count matrices for spliced and unspliced reads. Next, the count matrices were size-normalized to the median of total molecules across cells. The top 3000 highly variable genes are selected out of those that pass a minimum threshold of 10 expressed counts commonly for spliced and unspliced mRNA. For velocity estimation, we use the default procedures in scVelo (n_neighbors=30, n_pcs=30). In consideration that the assumptions of a common splicing rate and the observation of the full splicing dynamics with steady-state mRNA levels were often violated, we used the function recover_dynamics, a likelihood-based dynamical model, to break these restrictions. Finally, the directional flow is visualized as single-cell velocities or streamlines in the UMAP embedding with the Seurat cluster annotations.

**Connectivity of cell clusters**. To identify potential developmental relationships of cell clusters in patient #4 and #6, we utilized the partition-based graph abstraction (PAGA)[38] to estimate any potential developmental relationships among the three prostate lineages. The computations were performed on the same subset of variable genes as for clustering, using the default parameters.

**Identification of epithelial gene meta-programs**. Transcriptional programs were determined by applying NMF as previously described[46,47]. Analysis was performed for the epithelial cells only. We set the number of factors to 10 for each patient. For each of the resulting factors, we considered the 30 genes with the highest NMF scores as characteristics of that given factor (Supplementary Data 6). All single cells were then scored according to these NMF programs. Hierarchical clustering of the scores for each program using Pearson correlation coefficients as the distance metric and Ward's linkage revealed four correlated sets of programs with our focus.

**SCENIC**. In order to further investigate the gene regulatory networks (GRNs) in process of NED, we applied SCENIC[52] workflow to reconstruction the GRNs. The input matrices for SCENIC of every single sample was the corrected UMI counts in "SCT assay" of Seurat, in which we removed the variation of mitochondrial mapping percentage. For the combined sample (epithelial cells of 6 patients), Combat[71] was run to correct for "patient of origin" as source of batch effect. Following the standard procedure of SCENIC, we used GENIE3 (for single sample) and GRNBoost (for combined sample) to identify potential TF targets. Besides, the activity of each regulon in each cell is evaluated using AUCell, which calculates the Area Under the recovery Curve, integrating the expression ranks across all genes in a regulon. Finally, we used the default "AUCCellThreshholds" for each regulon as the threshold to binarize the regulon activity scores and created the "Binary regulon activity matrix". The motifs database for Homo sapiens was downloaded from the website https://pyscenic.readthedocs.io/en/latest/.

**Bulk dataset analysis**. Bulk-transcriptomic data were collected from Morrissey et al. (GEO:GSE126078)[30], Beltran et al. (https://www.cbioportal.org/study/summary?id=nepc_wcm_2016)[7], and Charles L. Sawyers et al. (https://github.com/cBioPortal/datahub/tree/master/public/prad_su2c_2019)[51]. To estimate the correlation of the P1, P2, and P4 meta-program with NED, we first defined an NE score by gene set variation analysis (GSVA)[72,73] for every sample in these bulk RNA-seq data, and the NE markers we used are listed in the materials. Then, we filtered cell cycle-related genes from the gene list of the three meta-program and performed Pearson correlation coefficient analysis of the remaining genes.

**Tissue microarrays**. Tissue specimens from 297 patients who underwent radical prostatectomy were collected for the construction of tumor microarrays (TMAs), and then the specimens were cut into 5-μm-thick sections using a standard microtome. These tissue cores were assessed by uropathologists to determine tumor stages according to the haematoxylin and eosin staining results (Supplementary Data 4).

**Immunohistochemistry (IHC) and immunofluorescence (IF)**. Formalin-fixed and paraffin-embedded tissue sections (5 μm) were deparaffinized and rehydrated. Antigen retrieval was carried out using 10 mM sodium citrate (pH 6.0) in a microwave oven. For DAB staining, endogenous peroxidase activity was blocked with 0.3% hydrogen peroxide for 10 min and 5% BSA in PBS for 1 h. Slides were incubated overnight at 4 °C with a primary antibody, which was followed by incubation with an HRP-linked secondary antibody (CST) at room temperature (30 min). Diaminobenzidine (DAB) was used as chromogen, and the sections were counterstained with haematoxylin. For immunofluorescence staining, the sections were washed with PBS and transferred to a blocking solution (10% normal donkey serum in PBS) for 1 h at room temperature. After blocking, specimens were incubated overnight at 4 °C with diluted primary antibodies. The next day, slides were washed with PBS three times for 10 min each, and then they were incubated for 1 h at room temperature with secondary antibodies conjugated to Alexa-488, -555, or -647, which were diluted with PBS containing 1% normal donkey serum (1:1000). Then, the secondary antibody was rinsed, and the slides were washed three times with PBS before being mounted with Vector Shield mounting medium containing DAPI (Vector Laboratories, H-1200).

**Image acquisition**. IF images were acquired using a Zeiss LSM 710 confocal microscope and were processed by ZEN Imaging Software. IHC images were acquired using an Olympus BX53 System Microscope.

**Primary antibodies**. The following antibodies were used in these studies: anti-SOX2 (Abcam, ab236557), anti-AR (Abcam, EPR1535(2)), anti-Cytokeratin 5 (Abcam, ab52635; For IHC), anti-CK5 (Biolegend, 905904; For IF), anti-SYP (Cell Signaling Technology, #36406), and anti-K18 (ProteinTech, 66187-1-Ig).

**Statistics and reproducibility**. Statistical analysis was performed using R (version 3.6.0) and GraphPad Prism (version 8). Wilcoxon rank-sum tests were used in this study and are described in each figure. Detailed descriptions of statistical tests are specified in the figure legends.

**Reporting summary**. Further information on research design is available in the Nature Research Reporting Summary linked to this article.

## Data availability
The scRNA-seq data were deposited in the NCBI Gene Expression Omnibus (GEO) database under accession number GSE137829. Source data underlying plots shown in figures are provided in Supplementary Data 1. All relevant data are available from the authors upon reasonable request.

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

## Acknowledgements
This work was supported by funds to W.Q.G. from Ministry of Science and Technology of the People's Republic of China (2017YFA0102900), National Natural Science Foundation of China (81872406 and 81630073), KC Wong Foundation, and the Shanghai Young Eastern Scholar Funds (QD2018021) to J.W.

## Author contributions
B.D. and W.Q.G. conceived and designed the study. J.M., W.L., and H.L. performed scRNA-seq analysis. Z.J., M.Z., X.C., J.M.W., and Y.W. carried out data analysis. H.H.Z., C.W.C., and Y.F. provided technical help. L.F., Y.Z., and J.P. gave advice for analysis and J.W. and W.X. wrote the manuscript with input from all authors. W.Q.G. supervised the project.

## Competing interests
The authors declare no competing interests.
