## [Peer Review File · Communications Biology]

Reviewers' comments:

Reviewer #1 (Remarks to the Author):

The study by Dong et al describes the molecular characterization - at the single cell level - of human patient samples from neuroendocrine prostate cancer (NEPC). Many advanced prostate cancers progress to NEPC following treatment failure and, to this Reviewer's knowledge, NEPC has not been well characterized at the single cell level. Therefore the study is important and timely. The study reports single cell data from 6 patients and the major conclusion is that NEPC has epithelial and more specifically luminal features. This will be important information to the community, and an important database resource.

Comments for suggested improvements:

1. The introduction could use some work -- it is a bit circular - starting with NEPC and then going to CRPC and then back -- it is also long and does not exactly represent the literature accurately.
 2. How were the patients chosen? There is quite a bit of heterogeneity among the patients.
 3. The logic for calling the NEPCs epithelial is not clear (Fig. 1 and discussion of)
 4. Figure 2 -- not clear what the data is showing (not well labeled)
 5. The NE score should be better described and supported - seems arbitrary
- The model - figure 8 - adds a lot -

Reviewer #2 (Remarks to the Author):

In this manuscript, the authors prepared single cells from biopsy tissues of 6 prostate tumors and performed single cell RNA-Seq analysis. From analyzing the sequence data in different ways, the authors report that the neuroendocrine tumor cells express both neuroendocrine cell markers and luminal like epithelial markers. Through lineage trajectory analysis, the authors suggested that focal neuroendocrine differentiation may originate from luminal like tumor cells rather than basal cells. By immunostaining prostate tumor TMAs, the authors identified 102 neuroendocrine prostate tumors, with 81% of these tumors were K18+/K5-/SYP+ and 13% were K18-/K5-/SYP+. Based these results, the authors suggested that neuroendocrine differentiation was primarily presented as a luminal feature. Through complicated data analysis, the authors also suggested a neuroendocrine differentiation gene signature. Overall, this is a very interesting study that identifies neuroendocrine prostate cancer cells at a single cell resolution level. This study also provides new gene expression signatures for neuroendocrine prostate cancer cells, which may help to explore new players and mechanisms that drive neuroendocrine tumor cell differentiation.

Major comments:

1. The authors used an approach to compare the gene expression profile of neuroendocrine tumor cells with that of basal and luminal epithelial cells. Because most neuroendocrine tumor cells did not express basal cell markers but express luminal epithelial cells, the authors concluded that neuroendocrine tumor cells are trans-differentiated from luminal tumor cells, but not basal cells. This logic and assumption to compare neuroendocrine cell markers with basal cell markers may be incorrect. Basal cells contain adult stem cells or progenitor cells of both luminal epithelial cells and neuroendocrine cells. Once these progenitor cells differentiate into either type of cells, the derived cells are no longer express basal cell markers. If a progenitor cells are transformed, it is possible to produce both luminal cancer cells and neuroendocrine tumor cells with or without luminal epithelial cell markers. Only RNA sequencing data at one time point is unable to figure out whether a neuroendocrine tumor cell with luminal marker is differentiated from a common progenitor cells or trans-differentiated from an existing luminal type tumor cell.

2. From the TMA immunostaining, the authors indeed identified 13% of K18-/K5-/SYP+ neuroendocrine prostate tumors out of all SYP+ tumors. This subtype does not express either

luminal or basal epithelial markers and therefore, the 6 samples assayed by single cell sequencing did not cover this subtype. This suggests that the conclusion that all neuroendocrine tumor cells express luminal epithelial marker genes obtained from a limited number of assayed samples is not very well supported.

Specific comments:

3. As the authors already mentioned in the manuscript, the number of samples analyzed is limited.

4. In figure 3C, it seems some sections of the same patient are adjacent sections and others are not. It is better to use serial sections for immunostaining performed with different antibodies. In particular, immunostaining for luminal cell markers such as CK18 should be performed with these sets of tissue sections to validate the s.c. RNA sequencing data. Importantly, the authors should pay attention to examine whether the SYP+ NE cancer cell cluster detected in patient #2 expresses luminal cell markers.

5. When the authors describe results for K18, K5 and SYP immunostaining, they claim that there was no SYP+/K18-/K5+ NE cells. This was not true, because the lower panels of Figure F clearly showed one SYP+/K18-/K5+ cell. Actually, the other cell in the same panel that was indicated as K18+/K5+/SYP+ does not show clear K18 staining signaling.

6. Some genes such as SOX2 and EZH2 were identified by this study as genes that are specifically expressed in the neuroendocrine tumor cells. However these genes are known to be more broadly expressed.

Reviewers' comments:

Reviewer #1 (Remarks to the Author):

The study by Dong et al describes the molecular characterization - at the single cell level - of human patient samples from neuroendocrine prostate cancer (NEPC). Many advanced prostate cancers progress to NEPC following treatment failure and, to this Reviewer's knowledge, NEPC has not been well characterized at the single cell level. Therefore the study is important and timely. The study reports single cell data from 6 patients and the major conclusion is that NEPC has epithelial and more specifically luminal features. This will be important information to the community, and an important database resource.

Reply: We appreciate this positive feedback from the reviewer that our manuscript provides a valuable single-cell resource for a better understanding of the cellular basis of neuroendocrine prostate cancer (NEPC). We have made several revisions, as suggested by the reviewer.

Comments for suggested improvements:

1. The introduction could use some work -- it is a bit circular - starting with NEPC and then going to CRPC and then back -- it is also long and does not exactly represent the literature accurately.

Reply: We thank the reviewer very much for the careful reading of our manuscript. As suggested, we have recomposed the introduction in line 54-80.

2. How were the patients chosen? There is quite a bit of heterogeneity among the patients.

Reply: As we focused our study on the NEPC patients and focal NED can be more frequently detected in patients with advanced prostate cancer undergoing ADT than in primary prostatic adenocarcinoma¹⁻⁴, we thus only chosen samples from CRPC patients. In addition, we preferentially selected patients whose circulating PSA level was lower than 20 ng/ml, indicating a higher likelihood of having NED in these patients. We have represented this information in the Methods.

3. The logic for calling the NEPCs is not clear (Fig. 1 and discussion of)

Reply: We thank the reviewer for pointing this out. To call the NE cells in each individual sample, we estimated the expression levels of 12 well-known NE markers (ASCL1, CHGA/B, FOXA2, SIAH2 et al) using Seurat AddModuleScore function. We have now described the detail of calling NE cells in both Results (line 130-133) and Method part (line 489-498). Of note, as reviewer #2 suggested that two NE markers (SOX2 and EZH2) are not specifically expressed in NE cells, we have now removed these two genes from the NE gene set and obtained consistent results with the original analysis.

In addition, we also showed expression levels of cell type markers including NE gene set in heatmap in Figure 3, which is also consistent with the NE score analysis that there are four patients in our datasets have NE cells.

4. Figure 2 -- not clear what the data is showing (not well labeled)

Reply: We thank the reviewer for pointing this out. We have now added more clear labels in Figure 2 and provided more detailed information in the figure legend.

5. The NE score should be better described and supported - seems arbitrary
The model - figure 8 - adds a lot -

Reply: We have now provided detailed pipeline of NE score analysis in the methods part. We used Seurat AddModuleScore function to evaluate cell identity based on a given gene set⁵⁻⁷. The NE gene set for defining NE phenotype were curated from the published papers, consist of 12 specific NEPC markers (such as SYP, CHGA/B, ENO2, et al).

Reviewer #2 (Remarks to the Author):

In this manuscript, the authors prepared single cells from biopsy tissues of 6 prostate tumors and performed single cell RNA-Seq analysis. From analyzing the sequence data in different ways, the authors report that the neuroendocrine tumor cells express both neuroendocrine cell markers and luminal like epithelial markers. Through lineage trajectory analysis, the authors suggested that focal neuroendocrine differentiation may originate from luminal like tumor cells rather than basal cells. By immunostaining prostate tumor TMAs, the authors identified 102 neuroendocrine prostate tumors, with 81% of these tumors were K18+/K5-/SYP+ and 13% were K18-/K5-/SYP+. Based these results, the authors suggested that neuroendocrine differentiation was primarily presented as a luminal feature. Through complicated data analysis, the authors also suggested a neuroendocrine differentiation gene signature. Overall, this is a very interesting study that identifies neuroendocrine prostate cancer cells at a single cell resolution level. This study also provides new gene expression signatures for neuroendocrine prostate cancer cells, which may help to explore new players and mechanisms that drive neuroendocrine tumor cell differentiation.

Reply: We appreciate the reviewer to acknowledge the novelty the single-cell data. We have addressed most concerns, as described below.

Major comments:

1. The authors used an approach to compare the gene expression profile of

neuroendocrine tumor cells with that of basal and luminal epithelial cells. Because most neuroendocrine tumor cells did not express basal cell markers but express luminal epithelial cells, the authors concluded that neuroendocrine tumor cells are trans-differentiated from luminal tumor cells, but not basal cells. This logic and assumption to compare neuroendocrine cell markers with basal cell markers may be incorrect. Basal cells contain adult stem cells or progenitor cells of both luminal epithelial cells and neuroendocrine cells. Once these progenitor cells differentiate into either type of cells, the derived cells are no longer express basal cell markers. If a progenitor cells are transformed, it is possible to produce both luminal cancer cells and neuroendocrine tumor cells with or without luminal epithelial cell markers. Only RNA sequencing data at one time point is unable to figure out whether a neuroendocrine tumor cell with luminal marker is differentiated from a common progenitor cells or trans-differentiated from an existing luminal type tumor cell.

Reply: We thank the reviewer for this insightful suggestion and agree that “Only RNA sequencing data at one time point is unable to figure out whether a neuroendocrine tumor cell with luminal marker is differentiated from a common progenitor cells or trans-differentiated from an existing luminal type tumor cell”. Indeed, we have discussed that our single-cell data can’t exclude the possibility that basal stem cells can serve as the cell of origin of both PCA and NEPC and suggested that future single-cell studies of serial tumor samples from individuals will be needed in principle to map the cellular dynamic involved in NED process as much as possible (line 403-407). We also represented a schematic illustration of tumor evolution toward the neuroendocrine phenotype, proposing that both basal and luminal cells can be the tumor initiating cells. However, based on our TMA data, we considered that direct basal-NE transdifferentiation is likely rare in human prostate cancer, but rather luminal-NE transdifferentiation is fundamentally responsible for phenotypic transition from prostate adenocarcinomas towards NEPC (line 387-396).

2. From the TMA immunostaining, the authors indeed identified 13% of K18-/K5-/SYP+ neuroendocrine prostate tumors out of all SYP+ tumors. This subtype does not express either luminal or basal epithelial markers and therefore, the 6 samples assayed by single cell sequencing did not cover this subtype. This suggests that the conclusion that all neuroendocrine tumor cells express luminal epithelial marker genes obtained from a limited number of assayed samples is not very well supported.

Reply: We thank the reviewer for this comments and have made revisions to highlight that the 13% of K18-/K5-/SYP+ neuroendocrine prostate tumors out of all SYP+ tumors didn’t be covered in our single-cell samples and used more softened conclusion in the single-cell results part (line 262-264). Overall, the TMA immunostaining data generally support single-cell result, given that most SYP+ tumor cases (81%) expressed a luminal phenotype.

Specific comments:

3. As the authors already mentioned in the manuscript, the number of samples analyzed is limited.

Reply: we agree with the reviewer that more samples analyzed can strengthen the conclusion of this study. However, it should be noted that the development of castration-resistant prostate cancer (CRPC) after androgen deprivation therapy need to wait for several months to years and the unique NEPC cases account for only about 10% of the total CRPC⁸, which brought a challenge for collecting abundant NEPC samples in certain time.

4. In figure 3C, it seems some sections of the same patient are adjacent sections and others are not. It is better to use serial sections for immunostaining performed with different antibodies. In particular, immunostaining for luminal cell markers such as CK18 should be performed with these sets of tissue sections to validate the s.c. RNA sequencing data. Importantly, the authors should pay attention to examine whether the SYP+ NE cancer cell cluster detected in patient #2 expresses luminal cell markers.

Reply: We appreciate this advice and have now added CK18 IHC data in Figure 3C. Unfortunately, because the prostate cancer biopsy samples were very small and limited, we have no more sections of patient #5 for CK18 staining. Furthermore, co-staining analysis of CK18 and SYP in section of patient #2 (supplementary fig. 4) have been performed and confirmed that NEPC cells in fact also manifest luminal phenotypes (line 195-197).

5. When the authors describe results for K18, K5 and SYP immunostaining, they claim that there was no SYP+/K18-/K5+ NE cells. This was not true, because the lower panels of Figure F clearly showed one SYP+/K18-/K5+ cell. Actually, the other cell in the same panel that was indicated as K18+/K5+/SYP+ does not show clear K18 staining signaling.

Reply: Indeed, in the lower panels of Figure F, the expression of K18 is detectable but at relatively low expression levels. We thus grouped this case as the K18+/K5+/SYP+ case. In addition, as a contrast, we have performed this triple staining in adjacent non-malignant tissues and there are clear SYP+ cell with strong K5 expression but without K18 expression (shown in fig. B below). Thus, we feel confident to group this only case detected in tumor tissues as SYP+/K18+/K5+ cells (shown in fig. A below).

6. Some genes such as SOX2 and EZH2 were identified by this study as genes that are specifically expressed in the neuroendocrine tumor cells. However these genes are known to be more broadly expressed.

Reply: We agree with the reviewer that some NE marker genes such as SOX2 and EZH2 are not specifically in NE cells. To assess the effect of SOX2 and EZH2 on NE cell calling, we thus removed them and calculated NE index again, which showed consistent results that 4 patients display NE differentiation signature (shown in figure below). In the revised paper, we have removed these two genes in defining NE identity.

References

1. Nelson, E. C. *et al.* Clinical implications of neuroendocrine differentiation in prostate cancer. *Prostate Cancer Prostatic Dis.* **10**, 6–14 (2007).
2. Huss, W. J., Gray, D. R., Werdin, E. S., Jr, W. K. F. & Smith, G. J. Evidence of Pluripotent Human Prostate Stem Cells in a Human Prostate Primary Xenograft Model. **90**, (2004).
3. Vashchenko, N. & Abrahamsson, P. A. Neuroendocrine differentiation in prostate cancer: Implications for new treatment modalities. *Eur. Urol.* **47**, 147–155 (2005).
4. Berruti, A. *et al.* Chromogranin A Expression in Patients With Hormone Naïve Prostate Cancer Predicts the Development of Hormone Refractory Disease. *J. Urol.* **178**, 838–843 (2007).
5. Puram, S. V. *et al.* Single-Cell Transcriptomic Analysis of Primary and Metastatic Tumor Ecosystems in Head and Neck Cancer. *Cell* **171**, 1611-1624.e24 (2017).
6. Bowling, S. *et al.* Resource An Engineered CRISPR-Cas9 Mouse Line for Simultaneous Readout of Lineage Histories and Gene Expression Profiles in Single Cells II Resource An Engineered CRISPR-Cas9 Mouse Line for Simultaneous Readout of Lineage Histories and Gene Expression Profiles in Single Cells. *Cell* **181**, 1410-1422.e27 (2020).
7. Karthaus, W. R. *et al.* Regenerative potential of prostate luminal cells revealed by single-cell analysis. **505**, 497–505 (2020).
8. Small, E. J. *et al.* Clinical and genomic characterization of metastatic small cell/neuroendocrine prostate cancer (SCNC) and intermediate atypical prostate cancer (IAC): Results from the SU2C/PCF/AACR West Coast Prostate Cancer Dream Team (WCDT). 5019–5019 (2016).

REVIEWERS' COMMENTS:

Reviewer #1 (Remarks to the Author):

The authors have addressed my concerns.

Reviewer #2 (Remarks to the Author):

In the revised manuscript, the authors have addressed the issues raised by this reviewer from reviewing the initial version of manuscript. The quality of the manuscript has been further improved.

REVIEWERS' COMMENTS:

Reviewer #1 (Remarks to the Author):

The authors have addressed my concerns.

Re: Thank the reviewer for taking time to review our manuscript.

Reviewer #2 (Remarks to the Author):

In the revised manuscript, the authors have addressed the issues raised by this reviewer from reviewing the initial version of manuscript. The quality of the manuscript has been further improved.

Re: Thank the reviewer for taking time to review our manuscript.